# The 12 Rs Framework as a Comprehensive, Unifying Construct for Principles Guiding Animal Research Ethics

**DOI:** 10.3390/ani13071128

**Published:** 2023-03-23

**Authors:** Christiaan B. Brink, David I. Lewis

**Affiliations:** 1Centre of Excellence for Pharmaceutical Sciences, North-West University, Potchefstroom 2531, South Africa; 2School of Biomedical Sciences & Biological Sciences Teaching Innovation Hub, University of Leeds, Leeds LS2 9JT, UK

**Keywords:** animal research ethics, 12Rs framework, 3Rs, harm-benefit analysis, ethical principles, animal rights, animal welfare, scientific integrity, social values, responsible research

## Abstract

**Simple Summary:**

Animals have been used in research for various purposes, ranging from animal, agricultural, wildlife, environmental and medical sciences to education and training. The ethical and humane use of animals for scientific purposes is often referred to animal research ethics. The latter has matured substantially over the past decades, so that it has become a cornerstone for responsible conduct of research. Scientific studies using animals are diverse and specialized in nature, and the intricacy of animal welfare and other key ethical aspects to be considered has also become expansive and complex, requiring a simplified, birds-eye view model, here called the 12Rs Framework, to guide all stakeholders between key considerations and fostering ethical integrity. Furthermore, the framework takes into account local context, legal requirements, values, and cultures, rendering it universally applicable. In fact, the 12 Rs framework can be a useful tool to promote, facilitate and harmonize the ethical conduct in the use of animals for scientific purposes across the world, almost like a mind map to navigate a complex but essential arena of research ethics.

**Abstract:**

Animal research ethics and animal welfare in science have become progressively tightly regulated, and ethical integrity and scientific quality, as well as social responsiveness and responsibility have become key requirements for research to be approved, funded, published, and accepted. The multitude of factors to contemplate has in some instances not only become complex, requiring a team approach, but often perceived as confusing and overwhelming. To facilitate a process of simplistic yet comprehensive conceptualization, we developed the 12 Rs Framework to act as a mind map to guide scientists, oversight structures, and other stakeholders through the myriad of ethical considerations. It unfolds into three domains of twelve encompassing ethical principles, values, and other considerations, including the animal welfare, social values, and scientific integrity domains, whilst also recognizing the diversity of local context, legal requirements, values, and cultures around the globe. In the end, it can be seen as a unifying ethical framework to foster and promote animal research ethics.

## 1. Introduction & Background

Today, the care and use of animals for scientific purposes is increasingly well regulated in the Global North and increasingly so in the Global South. In fact, there has been a rising awareness amongst all stakeholders within the scientific community, and growing insight specifically regarding animal research ethics. This has led to new constructs being formulated over the years to facilitate new perspectives, including express values that have emerged and been refined, as well as principles, norms, and standards that have been set. Even though all these new developments have facilitated positive research practices, they simultaneously have led to scientists and other role players often being confused and overwhelmed. The myriad of new information and regulations, and the need and expectations to adhere to all these developments have rendered the environment complex. Many researchers, in fact, experienced it confusing or even annihilating to adhere to new and constantly developing, often vehemently implemented ethico-legal frameworks. It became apparent that there is indeed a multitude of matters to consider when planning, reviewing, approving, executing, managing, monitoring, and reporting on any animal study, or on any other activities involving the use of animals for scientific purposes. Accordingly, the need for a comprehensive, unifying framework of animal research ethics principles, to guide researchers throughout their engagement with animal research, has become increasingly important. Such a framework could also become a useful tool, almost as an educative birds-eye-view, to see and understand the bigger picture, thereby becoming a larger construct to assist scientists using animals to ensure the inclusion of all new the latest ethico-legal developments and to help them to navigate successfully and responsively though the research processes.

### 1.1. A Brief History and Contextualisation

Following atrocities committed against specifically humans in the name of medical science during 2nd World War, there has been, what can be called, a moral and ethical awakening in scientific communities. One can think of international conventions, such as the Nuremberg Code of 1947, the Declaration of Helsinki in 1964, and the Belmont Report in 1978, accompanied by numerous legislation and regulations in many countries across the globe, supplemented by ethical oversight structures and different forms of ethical guidelines. This awakening also filtered through to the recognition of animal sentience and their ability to suffer, their rights, and to increased consideration of the ethical use of animals for scientific purposes. This is not to say that our understanding of the need of animals to be protected against human exploitation is new. In fact, several religious and moral writings from Ancient Greece reflect on this, whereas first animal welfare laws were introduced in the United Kingdom in the 1800s [1]. In the 1960s, an increasing number of countries followed suite, including India [2], Kenya [3], and the USA [4]. Then, by the middle of the previous century, Russel and Burch introduced the principles of humane experimental technique, the 3 Rs (replace, reduce, and refine) for the humane care and use of animals in research [5], followed by the five freedoms (5fs) as articulated in the report of the Technical Committee lead by Roger Brambell [6].

### 1.2. The Use of Animals in Research

The first recorded use of animals for research was 4th Century BC in Ancient Greece [7]. Since then, the use of animals for scientific purposes has been associated with significant advances across many scientific and medical fields, yet it remains a contentious topic [8] (see reference for an interesting comparison of arguments for and against). In particular, there are schools of thought regarding the responsible use of animals as indispensable, and then also those opposing any and all kinds of use of animals in research, including animal right activists [9]. Regardless, animals, even more so higher order animals, are sentient beings, capable of not only experiencing positive emotions, such as joy and comfort, but also negative sensations and emotions, such as discomfort, stress, pain, and real suffering. This leaves us with an ethical dilemma that both using them and not using them have respective negative consequences, be it for animals, humans and/or the environment.

In this regard, animals are commonly used for scientific purposes by, for example, research and educational institutions, the pharmaceutical industry, and agricultural and wildlife/environmental groups. Their uses include several diverse applications, including the investigation of animal biology and behaviour to better understand the organisms and their ecological contexts (advancing fundamental knowledge), conservation biology, and the preservation of species, or as models of human or animal disease to better understand these conditions or to develop potential therapies, as well as for product testing, biological substance production or diagnosis, and educational and training purposes. Most of the animals in specifically medical research are used to model human conditions, and include the use of vertebrates, such as zebrafish, or mammals such as rodents, or even high order animals such as dogs or non-human primates. Animal modelling of a human condition adds another layer of complexity, and such instances require verification that these animal models indeed display a meaningful degree of validity [10,11], including, but not limited to, face validity (i.e., it would present with similar symptoms or visible traits as humans with the condition that is being modelled), predictive validity (i.e., it would respond similarly to interventions, such as treatments, as humans with the condition that is being modelled), and construct validity (i.e., it would present with similar biological underpinnings and/or deviations as humans with the condition that is being modelled).

The use of animals is often unavoidable to answer important scientific questions, since in situ or in vitro models cannot fully replicate the complex biology and interactions of physiological systems or psychological processes. In fact, deduced assumptions about multifaceted whole organisms from results obtained using reduced in situ or in vitro models would be an oversimplification of reality (i.e., reductionism), rendering inaccurate conclusions. On the other hand, the use of humans for experimentation is not always possible, since the risk of harm or actual harm may render such investigations unethical, rendering such scientific experimentation not feasible. In such instances, the use of animals for scientific purposes is often considered by many as a more plausible and ethically acceptable (to humans) alternative. Yet it still leaves us with an ethical dilemma, namely that animals are sentient beings, able of experiencing discomfort, stress, pain, and real suffering, yet we use them to improve human and animal health and well-being. Here, speciesism or prioritarianism comes into play, which have been debated, even strongly criticised, because it places more value on human than non-human animal life [12]. Again, this illustrates ethical dilemma as alluded to above.

### 1.3. Animal Research Ethics around the Globe

The Council for International Organizations of Medical Sciences and the International Council for Laboratory Animal Sciences (CIOMS-ICLAS) formulated 10 basic principles to adhere to when using animals for scientific purposes, in its International Guiding Principles for Biomedical Research Involving Animals [13]. The last principle acknowledges and states that “While implementation of these Principles may vary from country to country according to cultural, economic, religious and social factors, a system of animal use oversight that verifies commitment to the Principles should be implemented…” Indeed, religious, cultural, and other perspectives impact on how we see ethics in general. For example, Jewish and Christian views would emphasise responsible management of creation and compassion towards animals. Islamic views would place a strong emphasis on stewardship towards animals and strict instructions from the Qur’an not to hurt animals [14]. African views would strongly rely on the ‘Ubuntu’ concept, understanding society and nature as interdependent, where “I am what I am because of who we all are”, and where animals can be viewed as part of “our family” [15,16]. This interconnectedness is also seen in Hinduism and Buddhism, where nature is regarded as sacred [17]. However, regardless the different perspectives, there are golden threads throughout that allow for unified, shared values and conventions. In this regard, the focus moved specifically to shared ethical principles and conduct within the context of research, commonly being referred to as animal research ethics.

We can provide an illustrative, although not comprehensive, list to illustrate the point: In Australia, we have the Code for the Care and Use of Animals for Scientific Purposes, 8th ed. of 2013, in China, the Chinese National Guidelines (GB/T 35892-20181), in Europe, the consolidated EU Directive 2010/63, in India, the Committee for the Purpose of Control and Supervision of Experiments on Animals (CPCSEA), in South Africa, the South African National Standards on the Use and Care of Animals for Scientific Purposes (SANS 10386:2021 2nd ed.), and on the broader African continent, The Guidelines for the Establishment and Functioning of Animal Ethics Committees (Institutional Animal Care and Use Committees) in Africa, of 2019. Many countries’ guidance is based on Chapter 7 of the World Organisation for Animal Health (formerly OIE) Terrestrial animal health code: use of animals in research and education [18]. The African Union Commission’s “Agenda 2063-the Africa we want” envisions “A prosperous Africa based on inclusive growth and sustainable development”, where humane and ethical studies involving animals in research will play a critical role in realising this vision. This commentary capitalises on the rich cultural, religious, political, and socioeconomic diversity that is Africa, its existing ethical, legal, and regulatory frameworks for animal research, as well as other good practices from across the World. It offers solutions for the way forward to ensure the humane use of animal for scientific purposes, not only in Africa, but globally.

What remains apparent is that there is a world-wide need for and strive to ensure ethical conduct at all times during research involving the use of animals. This remains true regardless the purpose of the research, be it for the ultimate goal of improving human health, veterinary purposes, conservation, agriculture, or any other scientific purpose (see below).

### 1.4. The Landscape of Animal Research Ethics

Increasingly, across the world, the 3 Rs (see their definitions in the discussion of the 12 Rs Framework below) [5] are being implemented to optimise the ethical use of animals in research and to minimise their suffering [19]. The principles of the 3 Rs, as a central theme in animal research ethics, have served the scientific community well and have promoted the humane care and use of animals in research significantly.

However, as awareness and comprehension of animal research ethics by all those involved in the care and use of animals in research have progressed and matured, new developments have occurred in the way of concepts, values and guidelines, often reflecting the cultural, religious or societal views of animals of the community that created them. An important focus of these developments was the optimisation of the use of animals, thereby to refine the ethical use of animals in research. Some of these newly introduced developments were: the fostering of a *culture of care* for animals (and humans) [20], advocacy of the *five freedoms* and *five domains* for animals [18], *respect* for research animals, *rehabilitation*, the *justification* of the use of animals, the *identification*, *justification*, *severity categorisation*, and *mitigation* of actual or potentially *harmful interventions* to animals, associated active *welfare monitoring*, and defining of *humane endpoints*, as well as proper *harm-benefit analysis* of any proposed study [21,22,23]. Whereas the introduction of these developments enhanced the ethical and humane use of animals, it has also created a perceived myriad of concepts, values, principles, cultural influences and other considerations to be taken into account and resolved, very often leaving those involved in the care and use of animals in research, members of ethical oversight bodies and other stakeholders feeling overwhelmed and even confounded as how to ensure that they adhere to all the new developments.

Importantly, there has also been a growing understanding of the close link between the use of sound ethical principles and research quality, taking cognisance of the culture in which it is conducted. With poor quality and inappropriate animal welfare, scientific outcomes are modified and sometimes changed [24,25], rendering the data untrustworthy. It is increasingly acknowledged that high standards in research go hand-in-hand with the application of sound ethical principles, and that sound ethics inevitably leads to higher scientific standards in research. In fact, both the PREPARE Guidelines for designing and planning studies [26], and the ARRIVE Guidelines for reporting results [27] emerged from this understanding. Alack of sound planning and reporting of experiments using animals have too often led to poor reproducibility, reliability, and translatability of data [28,29], making all research efforts unnecessary and therefore unethical through the wasteful use of animals, misleading information, and the wasting of valuable resources.

From the scientific literature, it follows that the ethical guidelines, as well as legislative and regulatory frameworks governing animal research, are fragmented and patchy, across the world. Increasingly, at regional, national and global levels, the broad animal sciences research community, animal welfare bodies, Statutory and Regulatory Bodies, and other stakeholders are collaborating to create, share, and implement animal welfare and ethical guidance, standards, regulations, and legislation, as well as strengthen the hands of animal research ethics committees and their review processes. The collective aim is to enhance animal welfare and research ethics globally through harmonisation, taking into account the cultural, religious, political and socioeconomic diversity between nations, rather than imposing, inflexible, and restrictive standardisation of research ethics approaches and practices.

### 1.5. The Need for a Unifying Animal Research Ethics Framework

In light of the above, we propose a comprehensive, unifying framework of key principles to produce clarity and simplicity when contemplating all the essential ethical concerns, while conducting research with animals. Whereas it has become evident that the most widely accepted benchmark of the 3 Rs is of key importance and very valuable and useful in animal research ethics to foster and safeguard animal wellbeing, it is no longer adequate and all-encompassing of all the new developments. It was also not intended to reflect on the global cultural diversity of views of the place of animals in society. To address this deficit and as indicated before, the need for a more comprehensive and inclusive, yet workable, framework, applicable across nations and cultures, has become important to navigate researchers using animal in research through the landscape of ethical requirements. Previous attempts to do so have been criticised to have diluted the 3 Rs [30,31]. With the suggested 12 Rs framework, we will attempt include all essential and helpful principles, and to unify them in a singly construct/framework as a synergistic, globally inclusive approach, clarifying all key matters that need to be considered. It will thus render it a useful tool for all those involved in the care and use of animals in research, whether it is the investigator, the ethical oversight body, the animal caretaker, the veterinary or para-veterinary professional, or another stakeholder.

## 2. The 12 Rs Research Ethics Principles Framework for Research with Animals

### 2.1. Conceptualization of the Framework

The framework is envisaged of consisting of different research ethics constructs linked together to form a unifying, comprehensive framework of 12 Rs (see Figure 1) that guides researchers using animals, the ethical oversight bodies, the animal facility (e.g., laboratory/research or other housing facility) or site (e.g., farm, wildlife) management and staff, to be ethical in their care of these animals as well as during the conduct of research. It becomes the compass for guiding and safeguarding all-inclusive “Ethical Conduct of Research” in the use of animals for scientific purposes (referred to as “ethical integrity” in the Figure 1). It provides a birds-eye-view of key principles during the planning, review, approval, conduct and monitoring processes of research with animals, as well as the undertaking and reporting of the care and use of these animals. Others that might find the framework useful are those who need to provide the appropriate infrastructure and professional guidance for animal studies, as well as animal owners, research institutions, and other stakeholders, to ensure comprehensive ethical oversight, management, and response.

The research ethics principles are closely interlinked to an overarching umbrella of research integrity focusing on (1) the fostering of a climate of responsible conduct of research through providing support, organizational structures, effective communication and training opportunities, as well as (2) reporting and responding to irresponsible or questionable research practices through well-defined and described procedures and processes [32].

“Honesty” or truthfulness is the cornerstone of the 12 Rs framework. However, it is not merely assumed to result from good intent, but needs to be actively pursued, and even tested to protect against unintentional or hidden deceit and misguidance. It often requires checks and balances, including independent assessment and review.

The 12 Rs are embedded in three main R domains: the Animal Welfare Rs (AWRs), the Social Value Rs (SVRs), and the Scientific Integrity Rs (SIRs), as well as Domain Intersecting Rs (DIRs) (Figure 2). The 12 Rs each encompass an important research ethics principle.

### 2.2. The Animal Welfare Rs (AWRs)

The Animal Welfare Rs cover the three Rs principles of *replace, reduce,* and *refine* [5], as alluded to Figure 2. These principles were introduced to drive forward the implementation of the value of respect for animals and the importance of safeguarding animal welfare as central to animal research ethics. This value is increasingly being accepted and endorsed amongst the global research animal care and use community, as well as stakeholders and regulatory bodies around the world [19]. The progressive creation of 3 Rs centres and platforms around the world [33] has played a crucial role in promoting the adoption of these and other research ethics principles important to animal research ethics. To encourage uptake and implementation, these centres have made the information more accessible, providing educational resources, guidance, and other materials online as open access resources.

“**Replace**”, as first principle, implies an attempt to replace animals as sentient beings in experiments. This may play out as either absolute replacement by using humans, human tissue, or non-sentient alternatives to animals, or as relative replacement by using animal tissue after humane killing, or lastly by using less sentient animal alternatives, where such replacement will still allow achievement of the research objectives.

“**Reduce**”, as second principle if replacement in not possible, implies the implementation of the optimal number of animals to be used, being as few as possible without compromising scientific validity of the study.

“**Refine**”, as third principle, implies the optimisation of the experimental design and animal care and use (procedures) to minimise pain, suffering and distress. This could, for example, include environmental enrichment to optimise living conditions and quality of life, or anaesthesia to reduce pain. In other words, it is about effecting mitigations for animal care and use throughout its lifetime (cumulative effects of all factors on suffering, including the study-induced impact such as animal housing and experimental procedures), to minimise risk of suffering, without unduly compromising scientific quality.

Within this context, implementation of the animal welfare Rs (AWRs) should supersede scientific interest, but should not invalidate scientific interest.

### 2.3. The Social Value Rs (SVRs)

The Social Value Rs refers to *Respect, Responsibility*, and *Regulations* discussed and illustrated in in Figure 2.

“**Respect**” refers to our regard for the animal’s dignity and wellbeing, as well as for animal owner’s or community’s rights and confidentiality. In terms of animal dignity, important concepts that have emerged include a culture of care [20,34], the five freedoms-5Fs [6], the five domains [35] (conceptualised in 1994), and animal rights. These also impact on the application of, e.g., the AWRs, so that these principles cannot be viewed in isolation.

**“Responsibility**” refers to the obligation and accountability (see “*Reckoning*” below) of the investigator and research team towards the animals (regarding appropriate selection of animals and proper care–see the *“Note!”* below), researchers and technicians (regarding their biological safety), society (regarding the use of public funding and trust that science will be for the common good and that no unnecessary harms are being inflicted on animals), and the environment and ecology (regarding protection of its integrity and sustainability, as well as the protection of those who live/work in, or derive income from the land/environment).

“**Regulation**”, in the broader context, would refer to compliance with applicable national legislation and regulations (which are often vastly different across the globe, influenced by culture and beliefs), as well as with directives or standards, statutory and/or professional registration, authorisation, approval, accreditation, licence, permission (e.g., permits). It also refers to compliance with local policies and rules, local or universally adopted conventions, and even consideration of institutional rules, culture, and reputation.

**Note!** It is noted that in some contexts or countries choice is based on cost and availability, or even the lack of other non-animal resources. In such instances, it is crucial that proper consideration be given whether valid scientific information will be gained, whether proper measures can be implemented to protect animals against undue harm, and that benefit will indeed outweigh harm. In fact, invalid data resulting from inappropriate choices will contribute nothing to science, whilst harming animals and wasting resources. It may then be better to collaborate with scientists or groups that do have access to what is needed to achieve the study outcomes, or not use the animals at all.

### 2.4. The Scientific Integrity Rs (SIRs)

The Scientific Integrity Rs consists of *Reproducibility, Relevance*, and *Transferability*, as illustrated in Figure 2.

Scientific integrity refers to ensuring a sound research design and methodology that will result in reliable and valid data and outcomes that address the research objectives. The quality of the scientific data generated, and hence the trustworthiness of conclusions drawn, has been a highly contentious issue [36,37]. A survey undertaken by Monya Baker, published for Nature reported that 70% of respondents could not replicate other researchers’ pre-clinical studies, with 50% also unable to repeat their own studies [38]. This lack of reproducibility is unethical, animals have suffered unnecessarily, and human and financial resources have been wasted. As alluded to above, one example of how the community is addressing this is represented by the development of the PREPARE Guidelines for designing and planning studies [26] and the ARRIVE Guidelines for reporting results [27].

“**Reproducibility**” is about data obtained from a robust study design to address the research questions, appropriate animal numbers, and statistical analyses to warrant statistical validity and transparency in the reporting and sharing of information to allow the data to be reproduced and its generalisability to be established. Providing comprehensive descriptions of methodology (as per the ARRIVE Guidelines) and the deposition of raw data in open access repositories, or publishing additional data as supplementary data together with articles, may enhance transparency and ultimately reproducibility. Critically, this is explicitly linked to animal welfare, with many studies demonstrating that in-appropriate animal welfare can modify or even change data [39].

“**Relevance**” or significance of a study relates to its justification or value added, taking into account the real or expected/likely benefit to humans, animals or the environment. This is usually derived from the literature study and problem statement. Furthermore, it could relate to benefit for animal or human health, society, science, etc.

“**Transferability**”: or translatability, relates to that which is experimentally modelled, simulated or represented. For animal models of human disease, translatability may refer to the validity of the animal model in terms of its face, predictive and construct validity [40], and other criteria for validity [11,41,42]. However, for different fields of science, e.g., wildlife or environmental studies, transferability may refer to the extent that the experimental sampling and design would represent the larger population under investigation (i.e., generalisability) or other species or environments.

### 2.5. The Domain Intersecting Rs (DIRs)

The three main Rs structures illuded to in Figure 2 have intersecting domains leading to three further principles namely *Righteousness, Reliability*, and *Reckoning*.

“**Righteousness**” represents the intersection of animal welfare (SIRs) and social values (SVRs) in striving to be fair, good, noble, worthy, and earnest scientists and members of society, and hence respectable science. It takes into account our values for life and fairness towards animals, and marries that with globally diverse cultural, religious, and/or social values related to our own responsibility, respect and legislative requirements.

“**Reliability**” represents the intersection of scientific integrity (SIRs) and animal welfare (AWRs). It relates to the robustness, quality, trustworthiness, applicability across environments or contexts, and generalisability of the data generated and conclusions drawn, embodied in a culture of scientific quality and integrity. How it links SIRs and AWRs can be seen in, as mentioned above, that animals that are well and free from suffering and distress, yield more trustworthy data, as opposed to unwell or otherwise suffering animals. On the other hand, scientific integrity ensures that the data generated from the use of animals are indeed yielding some real benefit in exchange for any loss of animal health and wellbeing.

“**Reckoning**” represents the intersection of scientific integrity (SIRs) and social values (SVRs). It refers to accountability, by measures during the planning and execution and after conclusion of an animal study. It is therefore foremostly pro-active by putting measures in place during the planning of a study, including for animal welfare, human safety and environmental integrity, aspects of respect towards animals and humans, ensuring that the study is sufficiently resourced, as well as legal compliance. In the spirit of harmonisation, rather than standardisation, all should be implemented taking into account also the local context of legislation, culture and values. Then, it is about actively caring about animals and other stakeholders during the execution of the study, in particular by vigilant implementation of animal welfare monitoring and other safety measures, identifying any planned and unforeseen consequences of/during the study, appropriate care and/or rehabilitation of animals after the study, as well as post approval monitoring and reporting any adverse event and immediately adjusting as necessary. In this regard, within a culture of care (as mentioned under the principle of “Respect”) and to be practical, the demonstration of respect towards co-workers (i.e., caring) would include not to ask of others to do what you are not willing to do. Lastly there is accountability measures in final study reporting to the ethical oversight body, robust and complete reporting to the scientific community and/or community and any post-study analyses. Throughout, there are other checks and balances, e.g., offered by experts, professionals, appropriate infrastructure and its standard operating procedures, assurance of affordability to complete a study, sustainability, and the management of any real or perceived conflict of interest.

### 2.6. Harm-Benefit Analysis as Culminating Point of the 12 Rs

The “**Harm-Benefit Analysis**” forms the culmination and key ethical safeguard of the 12 Rs framework. This ethical safeguard, at the centre of the 12 Rs, indicates that above all the researchers will manage the harm done to the animals during research and ensure that the research does have benefit. It acknowledges that actual harm is done to animals by taking them out of their natural environment (contingent harms) and by subjecting them to experimental procedures (project related harms). It was first conceptualised in 1986 by Bateson [43] and later refined by Pound and Nicol [44] who described it as a “cornerstone of animal research regulation” and “key ethical safeguard”. Here, the benefit refers to value added to science, including to the betterment of human health or animal wellbeing, agriculture, conservation economy, or other scientific purposes, as alluded to above.

Analysis of *harm* involves that we take into account the nature of the harm (physical, psychological, etc.), the degree and likelihood of the harm, the cumulative nature of harms, the justification of the harm (why it is necessary), and the mitigation to minimise the harm induced [23,45]. Analysis of the *benefit* is deduced from the literature review and study objectives, the degree and likelihood of study success and benefit, as well as the nature and extent of the benefit (i.e., for animals, human health, society, environment, science, etc.) [23,45].

Finally, it is determined via independent and informed discussion and deliberation (by all stakeholders and ultimately by the ethical oversight body or Regulatory Authority) whether benefit outweighs harm. In fact, as an example, the European Directive Article 36 clause 2 (EU Directive 2010/63) requires a favourable project evaluation, referring to Article 38, in which clause 2(d) specifically mentions the requirement for a harm-benefit analysis (EU Directive 2010/63).

Ultimately, the harm-benefit analysis takes into account all of the 12 Rs and becomes the final frontier that will determine whether a study will withstand the test of overall ethical integrity. Appendix A that may be useful to further explain the 12Rs Framework and to be used as training tool, is Appendix A: An audio-visual executive summary of the essence of the 12 Rs Framework, useful also as training tool.

## 3. Incorporation of Moral Ethics Theories and Moral Principles in the 12 Rs Framework

The Kantian deontology and utilitarianism are the most prominent moral ethics theories adopted in animal research ethics [46], understanding that these viewpoints are all multifaceted, incomplete, and subject to criticism. The utilitarian view considers that consequences of action (per implication, pleasure or advancement, versus pain or detriment) will determine whether research is right or wrong. This view is very much aligned with the harm-benefit analysis that takes cumulative harms and overall benefit of an animal study into account [47]. The deontological view is based on duty, obligation, and rules, which could be interpreted as actions being good because they are aligned with conventions, legislation, and directives. The important question that is often raised is not merely whether animals are intelligent, but whether they are sentient, capable of suffering. It is often argued that animal have intrinsic value, worthy of equal moral consideration, in dispute of speciesism [12]. The latter would be in agreement with principlism and care ethics, but in contrast to an absolute application of prioritarianism. All of these moral ethical viewpoints are accommodated in non-absolute terms in the 12 Rs framework (compare Figure 2).

Furthermore, principlism would argue that the use of animals should respect autonomy, promote beneficence and safeguard against maleficence, as well as to uphold justice. It should also foster fidelity. ‘Respect’ here refers to the recognition of the intrinsic value of an animal and a regard of its dignity, such as by fostering a culture of care and the application of the five freedoms (to name two examples). ‘Beneficence and non-maleficence’ are incorporated by diligent application of the harm-benefit analysis, in particular via justification and the implementation of mitigation strategies. In this regard, Degrazia and Sebo [48] specified that animal research could be morally permissible only if it satisfies (a) an expectation of sufficient net benefit, (b) a worthwhile-life condition, and (c) a no unnecessary-harm/qualified-basic-needs condition. ‘Distributive justice’ would imply that, as one illustrative example, we do not overburden one animal with multiple harmful procedures, instead of having multiple animals undergo one procedure each. Considering it from the perspective of an individual animal, it is better that, e.g., 20 animals each have one injection rather than one animal having all twenty. The latter implies that refinement should take precedence over reduction. Lastly, the moral principle of ‘fidelity’ refers to honesty and scientific integrity.

## 4. Application of the 12 Rs Framework

The 12 Rs framework should be applied right through from conceptualisation, education and training, planning, approval, execution, monitoring, reporting, to concluding and eventual record keeping. It can be visualised as a *full circle*, with *dynamic consideration* and active application of the 12 Rs framework, and involves constant monitoring, learning and modification, and regular oversight and management (see Figure 3). When discussing the application of this framework, it is important to remain mindful that, although it may help us navigate through the various aspects to consider, it does not magically guarantee solutions to all problems. Again, harmonisation and not standardisation, should be considered, implying that one takes into account the local cultural, religious, political, and socio-economic diversity, as well as different stakeholders and the different stages of “scientific journey” as it changes over time. When applying moral ethics in real-life context, answers are frequently, almost habitually, confounded by a conflict between the ideals of different ethical principles. In this regard, a decision and action that would satisfy one ethical principle would often violate another, and this is commonly referred to as an *ethical dilemma*. In fact, in research ethics we are often confronted with the reality of such conflicts, so that thorough deliberation, the understanding of the facts, complexity, and context, as well as collective wisdom remain indispensable in finding best the resolutions to a problem.

Although, in the application of the 12 Rs framework, the investigator holds primary responsibility, the process requires many stakeholders to play a role in the oversight. These include ethical oversight panels (whatever they are called), regulatory bodies (statutory, professional and other), scientific oversight panels (whatever they are called, and whether separate from or integral to the ethical oversight body), animal care and use programmes (including policies and procedures, management and structures, staff, competency, infrastructure, etc.), strong and engaged Institutional leadership, and the involvement of independent animal use and care advancement bodies (animal welfare organisations, 3 Rs centres, etc.) and Society. To correctly implement the 12 Rs framework therefore implies a dynamic, ongoing, inclusive, and integrative process that involves all stakeholders.

The 12 Rs framework may become a useful tool for investigators in planning, executing, monitoring (and learning and modifying as required), reporting, and concluding studies. For ethical oversight bodies it may be a useful tool when developing its standard operating procedures, and for its members when doing reviews, approving and monitoring studies. In fact, all stakeholders, including relevant policymakers and regulatory bodies, should find the birds-eye view helpful.

For the sake of clarity, one illustrative example from a real-life scenario can be given. (1) Conceptualisation of a study using animals will follow from previous studies or encountering real-life problems, and/or engagement with literature. This leads to the formulation of a research question, study objectives, and hypotheses regarding potential outcomes. This dynamic process, as outlined above, is characterised by feedback loops. (2) Likewise, during study-specific education and training, the investigator is sensitised to ethical principles, gaining awareness, skills and broader understanding. (3) Thereafter the investigator will plan the study, considering all of the principles within the 12 Rs framework. In the process, there will be consultation with relevant experts, professionals, facility managers, and caretakers, as well as with literature, to better inform the proposal; hence, again, a feedback loop. (4) During a process of review by scientific peers and experts, and the ethical oversight body, new questions are posed, ensuring thorough consideration of matters that may have been missed. This prompts the researcher to go back and improve on what was first proposed. Eventually the amended proposed study may be approved. (5,6,7) As the researcher executes the study and monitors progress and animal wellbeing, unexpected practical obstacles and adverse events may emerge, prompting reporting, consultation, reconsideration, learning, and modification. This will again activate feedback loops between the investigator, scientific peers, literature and the ethical oversight body. (8) At last, when the study is concluded, successes and the actual harm-benefit ratio as it is materialised become apparent from final reflective reports and scientific reporting, as assessed by the ethical oversight body and peers. (9) Proper record keeping ensures quality control and future re-use, further analyses, and/or verification of data.

## 5. Synopsis & Conclusions

The 12 Rs framework of the ethical use of animals for scientific purposes is a comprehensive and unifying construct, being elucidating and useful for students, researchers, managers, ethics review bodies, policy makers, and regulatory bodies, guiding between the considerations and fostering overall ethical integrity. Since it allows for differences in local context, legal requirements, values, and cultures, it should be universally applicable. Using the framework, it is important to be familiar with and understand how to apply the various principles underlying the framework, as well as to understand the interdependency of the ethical domains (i.e., the animal welfare Rs, social values Rs and scientific integrity Rs, as well as the domain intersecting principles). The fundamental principle of honesty as a point of departure, and the value of a comprehensive harm-benefit analysis as concluding tool are all accentuated. It should be considered throughout the research process, from conceptualization and planning to reporting and conclusion. Furthermore, the 12 Rs framework incorporates various moral theories and moral principles in a synergistic manner, embracing also that we should seek answers collectively in systems that promote objectivity. Yet, it does not evade ethical dilemmas.

In the end, the 12 Rs framework can be a useful tool to promote, facilitate, and harmonize ethical conduct in the use of animals for scientific purposes across the world, almost like a mind map to navigate a complex but essential arena of research ethics.

## Figures and Tables

**Figure 1 animals-13-01128-f001:**
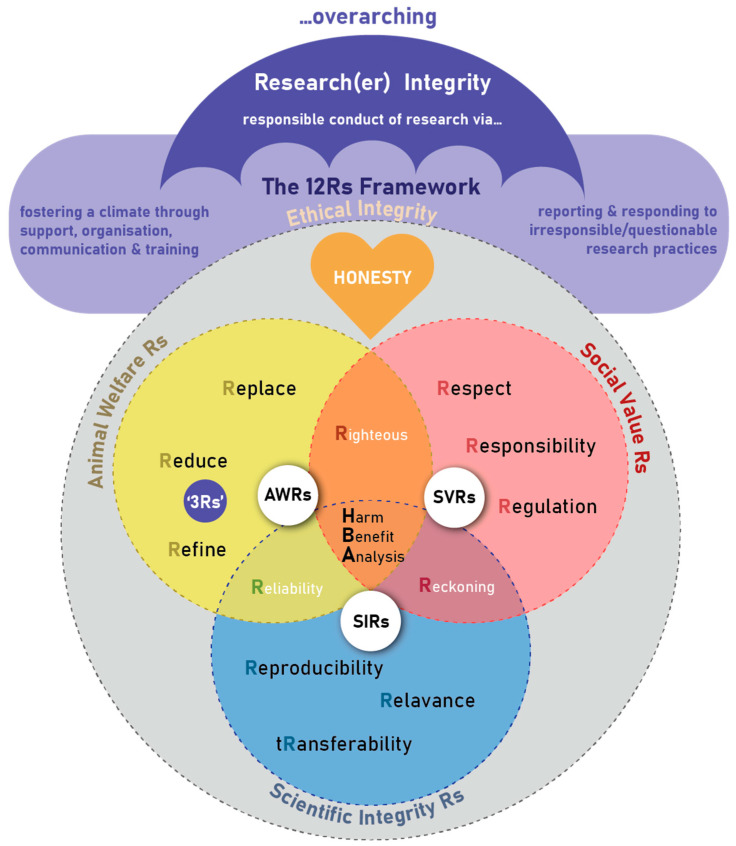
The 12 Rs framework for the ethical use of animals in research ethics: a birds-eye view.

**Figure 2 animals-13-01128-f002:**
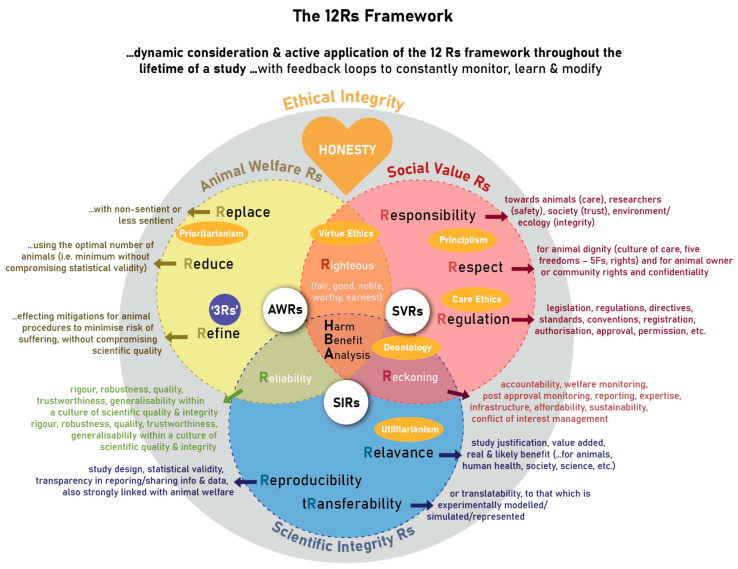
The more detailed exposition of the 12 Rs framework for the ethical use of animals in research ethics.

**Figure 3 animals-13-01128-f003:**
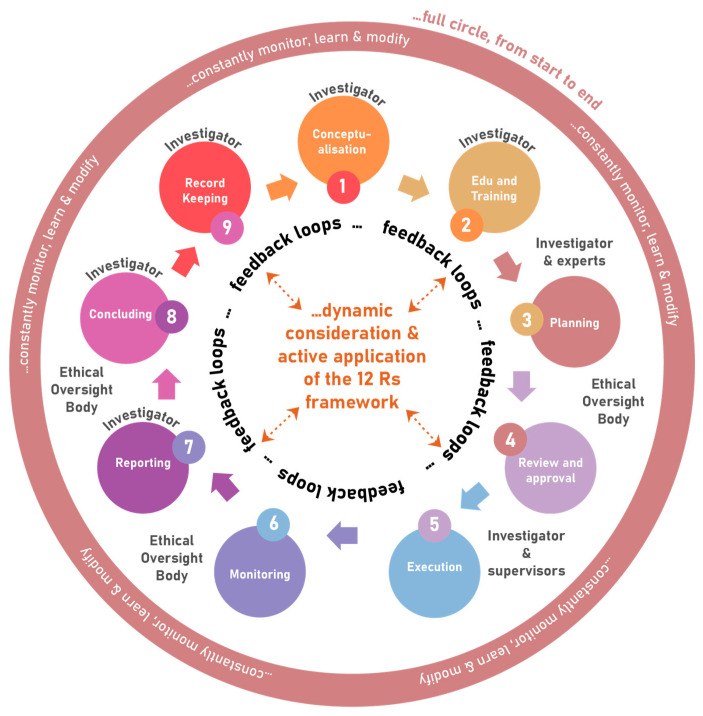
Application of the 12 Rs framework in studies using animals.

## Data Availability

Not applicable.

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
