# Peer review of "The 12 Rs Framework as a Comprehensive, Unifying Construct for Principles Guiding Animal Research Ethics"

_animals, 2023, doi:10.3390/ani13071128_

Round 1

Reviewer 1 Report

In this manuscript, Brink and Lewis are proposing a new framework for animal welfare, i.e. the 12R framework, as a further development of the 3R framework. 

The manuscript is well-written and the authors manage to present their ideas and new concepts clearly. I find the text as a very relevant contribution to the discussions and work in the animal welfare field and support this effort to diversify and develop the discourse in this field. 

Some minor comments:

Figure 2 should be enlarged to allow for an easier reading of the different "Rs". 

Rows 198 - 216: "It became clear" (row 198) and "it becomes clear" (row 202) is repeated, as well as "It has become evident" (row 214) and "it has become evident" (row 216).  

Section 4, row 451 - 488: The authors describe the potential future application of the 12 R framework in very general terms, but I would find it useful to include a couple of examples and real-life scenarios and describe how using the 12 R framework would be a useful tool in these cases.

Author Response

Comment 1

Figure 2 should be enlarged to allow for an easier reading of the different "Rs".

Authors’ response: We suggest that the layout of the figure be landscape in the final manuscript.  A high resolution has been uploaded as in JPG format.

Comment 2

Rows 198 - 216: "It became clear" (row 198) and "it becomes clear" (row 202) is repeated, as well as "It has become evident" (row 214) and "it has become evident" (row 216). 

Authors’ response: Thank you for pointing out these deficiencies.  The text has been amended in all three places indicated, now in lines 200, 205 and 217 of the revised manuscript.

Comment 3

Section 4, row 451 - 488: The authors describe the potential future application of the 12 R framework in very general terms, but I would find it useful to include a couple of examples and real-life scenarios and describe how using the 12 R framework would be a useful tool in these cases.

Authors’ response: Thank you for this valuable recommendation.  We have added a paragraph with an illustrative example from a real-life scenario in lines 501-.523 of the revised manuscript (just above paragraph 5).

Reviewer 2 Report

This is an interesting manuscript combining, in a very good way, all issues and trends related to the use of animals for research purposes. We believe that this manuscript will be an excellent "tool" for all those who are dealing with laboratory animal science and will probably start a useful dialogue between stakeholders. Some minor issues which need to be reviewed/corrected can be found below:

- Page 2, line 76: we propose to change the last words of the paragraph to  “followed by the five Freedoms (5Fs) as expressed by Brambell Committee”

- Page 2, line 91: Related reference is needed 

- Page 3, line 100: please write “EU Directive 2010/63”

- Page 2-3: We propose to change the row and put "1.3 the use of animals in research" before "1.2 animal research ethics around the globe"  

- Page 4, line 197 please add after ARRIVE Guidelines “for reporting results”

- Page 13 Line 537: the link for this reference has to be corrected because doesn’t work

- Page 14 line 547: a link for this reference has to be provided

- Page 14 lines  564 & 573: please add the access date

- Page 15 line 586: the link doesn’t work 

Author Response

Comment 1

Page 2, line 76: we propose to change the last words of the paragraph to “followed by the five Freedoms (5Fs) as expressed by Brambell Committee”

Authors’ response: Thank you, this gives clearer recognition to the contributors and enhances the informative value of the review.  We chose the working “…followed by the five freedoms (5fs) as articulated in the report of the Technical Committee lead by Roger Brambell”.

Comment 2

Page 2, line 91: Related reference is needed

Authors’ response: Thank you, we have added reference to P. Waldau’s paper (ref 17 in the amended manuscript)

Comment 3

Page 3, line 100: please write “EU Directive 2010/63”

Authors’ response: Thank you, this was corrected, now in line 148 of the amended manuscript..

Comment 4

Page 2-3: We propose to change the row and put "1.3 the use of animals in research" before "1.2 animal research ethics around the globe" 

Authors’ response: Thank you, this is indeed an excellent suggestion!  We applied this and paragraphs 1.2 and 1.3 were swopped accordingly in the amended manuscript.

Comment 5

Page 4, line 197 please add after ARRIVE Guidelines “for reporting results”

Authors’ response: Thank you, this was applied, now in both lines 199 and 343 of the amended manuscript.

Comment 6

Page 13 Line 537: the link for this reference has to be corrected because doesn’t work

Authors’ response: The DOI link to the article of Danayer (line 568-569, reference 10 in the amended manuscript) is as it appears in ScienceDirect (https://www.sciencedirect.com/science/article/pii/S2307502314000022) and in the PDF (https://www.sciencedirect.com/sdfe/reader/pii/S2307502314000022/pdf) of the manuscript.  Indeed when following the DOI link (https://doi.org/10.1016/j.nhtm.2014.08.001) it does not open (“not found”).  I have not been able to find the correct DOI link and kindly request the editorial team to assist with how to deal with this problem.

.Comment 7

Page 14 line 547: a link for this reference has to be provided

Authors’ response: Thank you, a link to the European Union’s Final Report has been included in the revised manuscript, being https://ec.europa.eu/environment/chemicals/lab_animals/pdf/report_ewg.pdf (accessed on 16 March 2023). Somehow EndNote does not include this a references of a report (as opposed to reference to a website), and it was added by hand.  I hope it is still included when reaching the editorial team.

Comment 8

Page 14 lines  564 & 573: please add the access date

Authors’ response: Thank you, the access dates for both Booth (ref 28, lines 619-621 in the revised manuscript) and Greeff (ref 32, lines 630-631 in the revised manuscript) were added.

Comment 9

Page 15 line 586: the link doesn’t work

Authors’ response: Thank you, the DOI link to Grimm’s article was corrected (ref 39, lines 649-650 in the revised manuscript).

Reviewer 3 Report

An excellent paper, clearly and logically argued, with an excellent, well reviewed and referenced overview of current ethical concepts in the field, synthesised into a very useful  framework that is likely to be of considerable benefit for animal (and human) welfare across multiple jurisdictions. 

 There are no changes that I would wish to make. (other to ask whether in line 40 it should read 'rising', rather than 'raising')?

Author Response

Comment 1

Ask whether in line 40 it should read 'rising', rather than 'raising')?

Authors’ response: Thank you for pointing out this error – it should read “rising”.  This was fixed, now in line 41 of the revised manuscript.